# Targeting Innate Immunity in Cancer Therapy

**DOI:** 10.3390/vaccines9020138

**Published:** 2021-02-09

**Authors:** Srikrishnan Rameshbabu, Brian W. Labadie, Anna Argulian, Akash Patnaik

**Affiliations:** Section of Hematology/Oncology, Department of Medicine, University of Chicago, Chicago, IL 60637, USA; srikrishnanrameshbabu@medicine.bsd.uchicago.edu (S.R.); blabadie@bsd.uchicago.edu (B.W.L.); annaargulian@uchicago.edu (A.A.)

**Keywords:** cancer immunotherapy, STING, NLRP3, tumor-associated macrophages, RIG-I, TLRs, CD40, NK cells, oncolytic viruses, pattern recognition receptors, innate immunity, cancer

## Abstract

The majority of current cancer immunotherapy strategies target and potentiate antitumor adaptive immune responses. Unfortunately, the efficacy of these treatments has been limited to a fraction of patients within a subset of tumor types, with an aggregate response rate of approximately 20% to date across all malignancies. The success of therapeutic inhibition of programmed death protein 1 (PD-1), protein death ligand 1 (PD-L1) and cytotoxic T-lymphocyte-associated antigen 4 (CTLA-4) with immune checkpoint inhibitors (ICI) has been limited to “hot” tumors characterized by preexisting T cell infiltration, whereas “cold” tumors, which lack T cell infiltration, have not achieved durable benefit. There are several mechanisms by which “cold” tumors fail to generate spontaneous immune infiltration, which converge upon the generation of an immunosuppressive tumor microenvironment (TME). The role of the innate immune system in tumor immunosurveillance and generation of antitumor immune responses has been long recognized. In recent years, novel strategies to target innate immunity in cancer therapy have emerged, including therapeutic stimulation of pattern recognition receptors (PRRs), such as Toll-like receptors (TLRs); the DNA sensing cGAS/STING pathway; nucleotide-binding oligomerization domain-like receptors (NLRs), such as NLRP3; and the retinoic acid-inducible gene-I (RIG-I)-like receptors (RLRs). In addition, therapeutic modulation of key innate immune cell types, such as macrophages and natural killer cells, has been investigated. Herein, we review therapeutic approaches to activate innate immunity within the TME to enhance antitumor immune responses, with the goal of disease eradication in “cold” tumors. In addition, we discuss rational immune-oncology combination strategies that activate both innate and adaptive immunity, with the potential to enhance the efficacy of current immunotherapeutic approaches.

## 1. Introduction

There is growing evidence that successful immune-mediated elimination of cancer requires coordination between the innate and adaptive arms of the immune system (Figure 1). Innate immune cells, such as dendritic cells (DCs), detect early cancers by a variety of mechanisms, including presentation of tumor-associated neoantigens or through sensing tumor-derived pathogen or damage-associated molecular patterns (PAMP/DAMPs) by pattern recognition receptors (PRRs) [1,2,3,4]. These mechanisms trigger proinflammatory programs and the release of proinflammatory cytokines, chemokines and type I interferons with accompanying DC maturation and trafficking to lymph nodes, where they engage the adaptive immune system and prime and activate antigen-specific T cells. T cells traffic and infiltrate the tumor bed through chemokine and cytokine gradients and mediate tumor elimination following interaction with their cognate antigen [5]. In addition, innate immune cells, such as macrophages and natural killer cells, contribute to tumor elimination through direct tumor cell killing by phagocytosis and cytotoxic mechanisms, respectively.

In recent years, evasion of antitumor immune response has been recognized as a hallmark of cancer and can be mediated by multiple mechanisms [6]. In the context of solid tumors, the absence of T cell infiltration on histopathologic analysis indicates one mechanism for evasion of antitumor immunity [7]. Gene expression analyses of the TME in solid tumors has identified genomic signatures which correlate with the presence or absence of T cell infiltration, referred to as “hot” and “cold” tumors, respectively [8,9,10]. More comprehensive classification has suggested four categories of TMEs: hot, altered-excluded, altered-immunosuppressed and cold [11]. These immune phenotypes have been observed to exist in a distribution, with certain cancer types having a higher proportion of “hot” immune phenotype, such as lung adenocarcinoma and clear cell renal cell carcinoma [12]. In addition, these immune phenotypes deploy distinct mechanisms to avoid immune-mediated elimination. For example, in response to T cell infiltration, “hot” tumors upregulate immune checkpoints, such as PD-1/PD-L1, within the TME, which directly suppress T cell effector mechanisms [13]. In contrast, “cold” tumors fail to generate spontaneous immune infiltration altogether through either “lack of antigenicity”, which results from defects in antigen processing or presentation, or “lack of immunogenicity” due to absence of tumor antigens capable of stimulating the immune system [14]. Of increasingly recognized significance, “cold” tumors prohibit T cell infiltration through orchestration of an immunosuppressive TME characterized by cell types, such as tumor-associated macrophages (TAMs) and myeloid-derived suppressor cells (MDSCs) [14,15,16].

“Hot” and “cold” tumor immune phenotypes carry prognostic significance, as solid tumors with T cell infiltration have been observed to have improved outcomes in multiple treatment settings [7,9,17,18,19]. To date, the success of therapeutic inhibition of PD-1/PD-L1 and CTLA-4 with immune checkpoint inhibitors (ICI) has been limited to “hot” tumors, while “cold” tumors have typically failed to benefit from ICI [9,19,20,21,22]. A large percentage of common malignancies, including prostate, breast and pancreatic cancers, are characterized as having “cold” TMEs and have historically not benefited from ICI. As such, there remains a critical unmet need to develop therapeutic approaches which drive innate immune activation and promote T cell infiltration in immunologically “cold” tumors. Herein, we review the roles of key innate immune cell types and discuss emerging therapeutic strategies to target these cell types to enhance antitumor immune responses, with the goal of disease eradication in “cold” tumors.

## 2. Key Cellular Components of Innate Immunity

Innate immunity comprises a diverse cadre of cell types which function as the body’s “first defense” against microbes. Each distinct cell type has a unique role; however, there exists overlap in function and cellular machinery, including expression of PRRs and proinflammatory response to detection of PAMPs/DAMPs. The expression of PRR and other innate immune pathways by innate immune cell type is summarized in (Table 1).

### 2.1. Dendritic Cells (DCs)

DCs are a family of antigen-presenting cells (APC) that perform critical functions in the initiation of antigen-specific immunity and tolerance. DCs express a diverse array of PRRs which, upon detecting PAMPs and DAMPs, lead to the upregulation of Major Histocompatibility Complex (MHC), costimulatory molecules required for T cell activation and CCR7 expression, the latter a key chemokine receptor that directs migration into tumor draining lymph nodes (TDLN) [23,24]. In TDLNs, DCs present tumor-associated antigens and generate antigen-specific CD8+ T cells as part of the cancer immunity cycle.

Dendritic cells derive from common myeloid progenitor cells and can differentiate into four primary DC phenotypes, which include conventional type I DCs (cDC1s), conventional type 2 DCs (cDC2s), plasmacytoid DCs (pDCs) and monocyte-derived DCs (MoDCs) [25]. cDC1s, characterized by BATF3 and IRF8 expression, are the most effective inducers of cell-mediated immunity due to their adept antigen processing and cross presentation. Multiple studies have demonstrated that BATF3-deficient mice are unable to eliminate immunogenic tumors or respond to immune-mediated therapies [4,26]. cDC2s have been found to enrich intratumoral CD4^+^ T cell density, which supports CD8+ T cell activity [27]. pDCs are a rare subtype of DCs that were initially termed interferon-producing cells (IPCs) due to their capacity to produce interferons and stimulate innate immunity when exposed to viral stimuli [28]. pDCs also function to directly regulate T cell activity; however, they have a comparatively blunted capacity to prime naïve T cells compared to conventional DC subtypes. More recent research suggests that pDCs may play a protumor role, as their infiltration into tumors is a poor prognostic factor in multiple cancers. The role of MoDCs in antitumor immunity is less clear; however, they have been shown to contribute to sustaining immunity following chemotherapy- or radiotherapy-driven cell death [24].

### 2.2. Macrophages

Macrophages are a type of myeloid cell that reside in healthy tissues throughout the body and perform critical functions to maintain tissue homeostasis and orchestrate innate immune responses [29]. TAMs exist in a continuum of polarization states between protumorigenic M2 macrophages and antitumorigenic M1 macrophages which correspond to dynamic gene expression programs [30,31,32,33]. In many cancer types, macrophages are driven to a M2 functional program which supports tumor growth, analogous to their role in tissue remodeling/wound healing, rather than anticancer immune activation. M2 TAMs mediate local immunosuppression via the production of IL-10 and TGF-β, suppression of T cell proliferation via extracellular arginine-depletion and enrichment of regulatory T cells via secretion of CCL2 [34,35,36]. Differentiation of a monocyte precursor to an M2 macrophage phenotype is promoted by hypoxia and immunosuppressive cytokines, such as IL-4, IL-10 and IL-13 [37,38]. In contrast, antitumorigenic M1 macrophages facilitate tumor control via multiple mechanisms, including phagocytosis and secretion of proinflammatory cytokines, such as IFN-γ, IFN-β and IFN-α [39]. Recent efforts in single-cell RNA sequencing of solid tumors have identified multiple subclusters of TAMs, which suggests that macrophage functionality exists across a continuum of states, and therefore, binary classification inadequately represents complex TAM phenotypes. For example, one study in pancreatic adenocarcinoma identified five subsets of TAMs each with distinct gene expression profiles which correlate with the diverse immunosuppressive or immune-stimulatory functions listed above [40,41]. Histologic analysis of clinical specimens demonstrated that tumor-associated macrophages (TAMs) are associated with worse overall survival [42]. In preclinical models, TAM depletion enhanced efficacy of radiation, chemotherapy and ICI [43,44,45].

### 2.3. Neutrophils

Neutrophils are circulating myeloid cells which function in the innate immune system’s response to bacterial infection. Neutrophils have been observed in high proportions within the immune infiltrate of many solid tumors, and elevations in both tumor infiltrating neutrophils (or tumor-associated neutrophils, TANs) and peripheral blood neutrophils associate with unfavorable outcomes [46,47,48]. Protumorigenic mechanisms of TANs include promotion of neoangiogenesis, tumor migration and invasion and local immunosuppression [49,50,51]. In contrast, TANs have been observed to exert antitumorigenic effects via direct tumor cell cytotoxicity, production of reactive oxygen species and secretion of proinflammatory cytokines [52,53,54]. The functional plasticity of TANs has led to a bipolar classification similar to that of TAMs with protumorigenic N2 TANs and antitumorigenic N1 TANs. Research has asserted that N1 TANs predominate in early tumor development. However, TGF-β, IL-10 and IL-6 signaling supports protumorigenic N2 TANs differentiation over time [55,56,57]. Despite recent advances, the tumor immunobiology of TANs remains largely under active investigation.

### 2.4. Myeloid Derived Suppressor Cells (MDSCs)

MDSCs are a heterogenous group of myeloid cells distinguished from other myeloid cell types by their predominantly immunosuppressive properties. Differentiation to MDSC phenotype is associated with chronic inflammation and low-level exposure to growth factors and inflammatory mediators responsible for normal maturation of myeloid cells [58]. Morphologically, MDSCs exist as mononuclear or polymorphonuclear subsets referred to as Mo-MDSC and PMN-MDSC, respectively. Compared to classical neutrophils and monocytes, MDSCs have an increased expression of immunosuppressive molecules, such as nitric oxide (NO) and IL-10, weaker phagocytic abilities and a higher expression of the immunosuppressive enzyme arginase-1 [59]. In many solid tumor types, MDSCs have been observed to suppress both innate and adaptive arms of the immune system and contribute to a “cold” TME [60,61]. MDSC infiltration into the tumor site correlates with increased cancer stage and tumor burden as well as worse prognosis [59,62]. These properties highlight the relevance of MDSCs to therapeutic strategies aimed at overcoming an immunosuppressive TME.

### 2.5. Mast Cells

Mast cells are granulated innate immune cells that reside in peripheral tissues and secrete a wide array of signaling molecules that facilitate tissue repair and local immune responses. The presence of mast cells in the tumor stroma of several solid tumors has been associated with poor prognosis. However, in breast cancer, the presence of mast cells was found to be a positive prognostic factor [63,64,65]. Precise localization of mast cells within the TME may also impact tumor development [66]. Molecules secreted by mast cells, such as vascular endothelial growth factor (VEGF) and matrix metalloproteases, have been observed to support tumor growth and metastatic potential through the promotion of angiogenesis and lymphangiogenesis and modification of the extracellular matrix [67,68]. Tryptase secreted by mast cells acts as an agonist of proteinase-activated receptor-2 (PAR-2) to further stimulate endothelial cell proliferation and has been associated with tumor cell migration [69]. Anticancer activities of tumor-associated mast cells include mediating tumor cell apoptosis through inflammatory mediators and the peroxidase system [70,71]. The comprehensive understanding of the function of mast cells in the TME remains an area of active investigation.

### 2.6. Natural Killer (NK) Cells

NK cells belong to a heterogenous family of innate lymphoid cells which lack genetically rearranged antigen receptors and do not require APC-dependent antigen presentation and selection for their cytotoxic activity. NK cells detect stress-induced molecules and altered or downregulated MHC class-I and exert perforin and granzyme-dependent antitumor cytotoxicity similar to CD8^+^ T cells [72]. In addition, NK cells are activated upon simultaneous binding of multiple receptors, including NKp46, NKG2D, 2B4, CD2 and DNAM, all of which are upregulated in the presence of cellular stress [72]. In addition to cytolytic functions, NK cells can also mediate Fas-ligand-induced target cell apoptosis [73]. It has been long known that NK cells exert early control of transformed cells, thereby serving an important role in tumor immunosurveillance [74]. Early work exploring the antitumor activity of NK cells revealed that NK cell depletion in MCA-induced fibrosarcoma led to significant tumor growth [75]. More recent studies have shown that the release of IFN-γ and chemokines CCL5 and CXCL1/2 by NK cells can potentiate adaptive immune responses through DC activation and induction of M1 macrophages [76,77,78]. Studies have also revealed that NKG2D activation stimulates macrophages and CD8^+^ T cells [79]. Histologic analysis of human pulmonary squamous cell carcinoma and adenocarcinoma specimens revealed an inverse correlation between NK cells and metastatic disease burden and mortality [80,81,82]. Taken together, the biological rationale and preclinical findings described above suggest a significant potential of NK cell-directed therapies in generating meaningful anticancer responses.

## 3. Activating the Innate Immune System in Cancer Therapy

Given the complex interplay of innate immune cells within the TME, targeting its key components could provide therapeutic benefits in cancer. There are several approaches to target innate immunity in cancer (Table 2). These include therapeutic stimulation of PRRs, such as Toll-like receptors (TLRs); the DNA sensing cGAS/STING pathway; nucleotide-binding oligomerization domain-like receptors (NLRs), such as NLRP3; and the retinoic acid-inducible gene-I (RIG-I) like receptors (RLRs) with synthetic agonists which resemble cognate PAMP/DAMPs (Figure 2). Direct therapeutic targeting of TAMs, NK cells and DCs are also being evaluated (Figure 3). Within each section, we review rational immune-oncology combination strategies that activate innate immunity, thereby enhancing the efficacy of current immunotherapies.

### 3.1. Toll-like Receptors (TLRs)

TLRs are highly conserved transmembrane and intracellular PRRs found in a variety of cell types and play a critical role in the detection of microbial pathogens by innate immune cells [83]. In humans, ten TLRs have been identified, and they are expressed by T cells, B-cells, APCs as well as many non-immune cells, including epithelial and endothelial cells. TLRs localize to two different regions of the cell: the plasma membrane and the endosome. TLR 1, 2, 5 and 6 are found specifically in the plasma membrane. TLR 3, 4, 7, 8 and 9 are found specifically in the endosomes, while TLR4 can signal at both locations [84]. TLRs located on the cell membrane recognize lipids and proteins, and TLRs located on the membrane of intracellular endosomes recognize nucleic acids. TLR signaling occurs through activation of adaptor proteins that enter the nucleus and regulate the expression of proinflammatory mediators. These adaptor proteins include MyD88, interferon regulatory factor 3 (IRF-3), NF-kB and activator protein-1 (AP-1) [85].

TLRs are expressed by both cancer cells and immune cells within tumors, and therefore, have pleiotropic effects on immunomodulation of the TME. When activated on immune cells, TLRs mediate a broad range of immunostimulatory effects that promote antitumor T cell responses [86]. Studies demonstrated that chemotherapy-induced DAMP release activated TLR4 and resulted in DC maturation and other immune activating effects [87]. TLR9 signaling has been shown to activate pDCs and result in the secretion of high levels of type I interferon [28]. TLR7 and TLR8 activation led to the reprogramming of tumor promoting M2 TAM phenotype to antitumor M1 phenotype [88,89]. On T cell populations, TLRs have diverse effects, including reduced suppressive function of T-regulatory (Treg) cells and enhancement of the survival, proliferation and cytokine production of CD8^+^ T cells [90,91,92]. Within cancer cells, TLR activation has been shown to trigger both apoptosis and cell survival [93]. In addition, constitutive activation of TLRs can lead to chronic inflammatory states which are associated with the recruitment of immunosuppressive cell types, such as myeloid derived suppressor cells (MDSCs), leading to tumor progression [94,95,96].

The diverse immunostimulatory functions of TLR activation provide ample rationale for therapeutic targeting for the treatment of tumors with a “cold” TME. Therapeutic administration of synthetic TLR agonists by intratumoral injection have revealed multiple anticancer effects in preclinical models. Activation of TLR9 by synthetic CpG-oligodeoxynucleotides was shown to revert resistance to PD-1 blockade by expanding multifunctional CD8^+^ T cells [97,98]. TLR9 agonist candidates SD-101 and CMP-001 were well tolerated in early phase clinical trials and demonstrated clinical activity in combination with anti-PD-1 treatment in melanoma and head and neck squamous cell carcinoma (HNSCC) [99,100]. These candidates are currently undergoing phase 2 trials for the treatment of solid tumors in combination with other forms of immunotherapy [NCT01042379, NCT04050085, NCT03007732, NCT03084640, NCT02554812, NCT03438318]. TLR7/8 agonist NKTR-262, in combination with the systemic CD122-biased IL-2 pathway agonist, bempegaldesleukin, promoted antigen presentation and CD8^+^ T cell infiltration in preclinical models [101]. This combination was well tolerated in early phase clinical trials and is currently being investigated in combination with anti-PD-1 therapy [NCT03435640]. Another TLR7/8 agonist, MEDI9197, was shown to activate pDCs and macrophages leading to interferon-α (IFN-α), IL-12 and IFN-γ release and subsequent antitumor T cell response and tumor regression in syngeneic murine models [102]. This molecule is currently being evaluated in early phase clinical trials [102]. Motilomod, a small molecule TLR8 agonist, is undergoing investigation in combination with nivolumab for HNSCC [NCT03906526].

Three TLR agonists are currently FDA-approved and in clinical use. Bacillus Calmette-Guerin (BCG), an attenuated strain of *Mycobacterium bovis* is a TLR2/4 ligand that is approved for treatment of superficial, non-muscle invasive bladder cancer [103]. In addition, the TLR4 agonist monophosphoryl A is approved as a vaccine adjuvant and TLR7 agonist imiquimod is approved for the treatment of genital warts and basal cell carcinoma [104,105,106].

### 3.2. cGAS/STING Pathway

The cGAS-STING pathway detects cytosolic DNA associated with viral infection and tumorigenesis. The cellular mechanisms by which the pathway mediates immune activation have been previously well described [107,108,109]. In brief, cGAS senses cytosolic DNA and activates Stimulator of IFN genes (STING) through synthesis of the cyclic dinucleotide, cyclic GMP-AMP (cGAMP) [110,111]. Upon activation at the endoplasmic reticulum and subsequent translocation to the golgi, STING activates IRF-3 and NF-kB transcriptional programs, resulting in the expression and release of type I IFN [112,113]. STING is expressed by multiple immune and non-immune cells, and its ability to sense tumor-derived DNA can be harnessed for cancer therapeutic purposes. In murine tumor models, STING-dependent cytosolic DNA sensing by tumor-resident DCs was found to induce type I IFN production and was required for CD8^+^ T cell infiltration and rejection of immunogenic tumors [108]. Intratumoral injection of STING agonists in preclinical models recapitulated these proinflammatory effects and induced profound tumor regression [114].

Therapeutic STING activation has been most successful with synthetic cyclic dinucleotides (CDNs) due to their structural versatility and ability to bind prevalent allelic variants in human STING [114]. Currently, there are multiple ongoing clinical trials with synthetic CDN STING agonists, which are comprehensively reviewed elsewhere [115]. MK1454 is undergoing clinical evaluation as monotherapy or in combination with pembrolizumab for the treatment of advanced solid tumors [NCT03010176, NCT04220866] [116]. ADU-S100/MIW815 is being investigated in phase II clinical trials in combination with ICI as a first line treatment for HNSCC [NCT03937141] [117]. Exploration of the dose-dependent effects of ADU-S100 revealed that while higher doses were more effective at clearing injected tumors, lower doses elicited IFN-γ-driven CD8^+^ T cell expansion and demonstrated synergy with ICI [118]. While most STING agonists have been administered via intratumoral injection, BMS-986301 is being evaluated for systemic intramuscular administration. Additionally, GSK3745417 is an intravenous STING agonist being evaluated alone and in combination with ICI in advanced solid tumors [NCT03956680, NCT03843359] [119]. A newly developed non-CDN amidobenzimidazole-based small molecule given intravenously displayed potent STING agonism and durable preclinical antitumor activity, thus representing an important step in adapting STING modulators for systemic administration [120]. Clinical evaluation of tolerability and safety is ongoing. Other emerging approaches include engineered liposomal nanoparticle packaging, ex vivo loading of STING into exosomes and bacterial modification to optimize treatment delivery [121].

The mechanistic underpinnings of the cGAS-STING pathway make STING agonists an attractive adjuvant to cancer vaccines. PancVAX is a vaccine composed of synthetic peptides and a STING agonist-based adjuvant, ADU-V19, which when used in combination with an OX40 agonist and anti-PD-1 therapy in preclinical models leads to significant tumor regression and improved survival [122]. The STINGVAX vaccine consists of a CDN ligand formulated with GM-CSF, which has potent antitumor activity as monotherapy across multiple murine models. It was shown to upregulate PD-L1 on the TME and resulted in combinatorial regression of tumors resistant to anti-PD-1 monotherapy [123]. STING agonists used in combination with chemotherapy and radiation therapy have been shown to amplify antitumor immune responses in both preclinical and clinical settings [124,125]. Interestingly, preclinical evaluation of poly ADP-ribose polymerase (PARP) inhibitors demonstrated a STING-dependent immune response which enhanced the efficacy of ICI [126].

Despite the encouraging framework for STING agonism in the treatment of cancer, some nuances with respect to context and dose-dependent effects are emerging. Recent research has demonstrated that cancers with high chromosomal instability (CIN) have increased levels of cytosolic DNA which contributes to endogenous cGAS/STING activation which, if sustained over time, can promote tumorigenesis, immune evasion and metastasis [111,127,128]. The existence of this phenotype represents an important consideration in developing rational strategies that incorporate STING agonists and pave the way towards mechanistic evaluation of intermittent vs. continuous STING pathway activation in generating durable antitumor responses.

### 3.3. Retinoic Acid Inducible Gene-I-like Receptors (RLRs) and RIG-I

RLRs are PRRs that detect cytosolic RNA physiologically in the context of viral infections [129]. The best studied RLRs are RIG-I, MDA5 and LGP2. RIG-I primarily binds shorter dsRNA, whereas MDA-5 interacts with longer fragments [130]. Upon binding of RNA, RLRs undergo a conformational change that exposes a CARD domain, which activates downstream effectors that promote the transcription factors IRF-1, IRF-3, IRF-7, NF-kb and IFN response elements, similar to the cGAS/STING pathway [127,128]. In addition, the CARD domain plays a role in inflammasome activation [131]. RIG-I mediates a wide range of immunostimulatory functions, including DC maturation, priming of T cells and enhancement of NK cell degranulation and cytolytic activity [132,133]. RIG-I also initiates programmed cell death (PCD) through both the intrinsic and extrinsic apoptotic pathways as well as pyroptosis, an inflammatory variant of PCD. The factors determining which form of cell death is driven by RIG-I is very context dependent and not clearly defined [132]. Furthermore, recent work has revealed that RIG-I is required for adequate response to anti-CTLA-4 treatment by inducing caspase-3-mediated tumor cell death, following which the tumor-associated antigens are processed and presented, leading to a robust CD8^+^ T cell-mediated adaptive immune response [134].

These proinflammatory mechanisms provide rationale for RIG-I as an innate immunotherapeutic target. Synthetic agonists specific to RIG-I or to RLRs more broadly, classified as RLR mimetics, are being investigated in multiple cancer types [135,136]. Intralesional RGT100 (MK-4621), a synthetic RIG-I agonist, demonstrated tolerability in a Phase I clinical trial and is now being studied as monotherapy or in combination with pembrolizumab for the treatment of advanced solid tumors [NCT03739138] [137]. In addition, synthetic stem-loop RNA (SLR) sequences that are highly specific for the RIG-I RNA binding pocket have been developed. SLRs represent a potent strategy for stimulating RIG-I due to precise structural optimizations and resistance to nucleases [138]. In vivo studies have shown that intratumoral delivery of SLR14 activated both NK cell and CD8^+^ T cell populations, resulting in significant antitumor effects [139]. A novel oral RIG-I agonist, SB9200, has demonstrated strong antiviral activity against resistant hepatitis C infection through induction of type I IFNs, and as such may have a role in future immunotherapeutic strategies [132,140].

Methods to stimulate RLRs more broadly include the use of poly-ICLC, a synthetic double-stranded RNA that activates TLR-3; MDA5; and, to a lesser extent, RIG-I [141,142]. In a pilot trial conducted with patients with refractory HNSCC and melanoma, an “autovaccination” strategy utilizing intralesional and intramuscular administration of poly-ICLC was well tolerated and led to the generation of antitumor T cell activation. This novel approach is now being further investigated in a phase II clinical trial [NCT02423863] [143].

### 3.4. CD40

Activation of an antigen-specific T cell response requires T cell receptor-mediated antigen presentation and interaction between costimulatory molecules, such as CD28 and the B7 family. CD40 is a well-characterized costimulatory molecule and member of the tumor necrosis factor (TNF) receptor superfamily and is highly expressed on APCs, including DCs, macrophages, B cells and many non-immune cells [144]. CD40 ligand (CD40L) is expressed by T cells and other non-immune cells. Upon cross-linking, CD40 triggers cell proliferation, upregulation of MHC and other costimulatory molecules and secretion of cytokines, including IFN-γ and IL-12a [145,146,147,148]. CD40-mediated CD8^+^ T cell activation has been shown to be independent of CD4^+^ T cells and innate immune sensors [149,150]. In addition, CD40 activates tumor-associated macrophages to the activated M1 phenotype. In one study, this resulted in the secretion of matrix metalloproteinases that modified tumor stroma and enhanced the effects of chemotherapy in a murine model of pancreatic adenocarcinoma [151,152].

Preclinical and early clinical experience has highlighted the efficacy of combining CD40 agonist with chemotherapy, radiation and peptide vaccines [153,154,155]. In an immunologically cold pancreatic ductal adenocarcinoma (PDA) murine model, treatment with a CD40 agonist alone yielded minimal response. However, the addition of chemotherapy resulted in a vaccine-like effect in which potent antigen-specific T cells were generated as a consequence of chemotherapy-induced cell death and antigen release [150]. Of note, CD40 agonism in combination with gemcitabine caused lethal hepatotoxicity in mice, ameliorated by giving CD40 agonist five days or more prior to chemotherapy [156]. This data established a rationale for a clinical trial of neoadjuvant and adjuvant selicrelumab, a CD40 agonist, in combination with nab-paclitaxel in resectable PDA [NCT02588443]. The addition of anti-PD-1 treatment further enhanced tumor regression, and a clinical trial in combination with anti-PD-1 therapy in PDA is underway and has observed encouraging early phase results [NCT02482168, NCT02304393] [157,158].

In recent preclinical studies utilizing murine models of non-immunogenic solid malignancies, triple therapy with a CD40 agonist, an anti-PD-1 antibody and a T cell activating vaccine stimulated macrophages and DCs reduced T cell exhaustion and generated effector memory CD4^+^ T cells [155]. In murine tumor models with high PD1^+^ T cells, CD40 agonism reversed T cell exhaustion and enhanced response to anti-PD-1 and anti-CTLA-4 immune checkpoint inhibition, suggesting a re-sensitization benefit for patients who experience resistance to ICI [159]. Furthermore, CD40 agonism and anti-CTLA-4 antibody achieved promising response rates and demonstrated increased T cell infiltration and activation in patients with melanoma [160]. From a clinical translational standpoint, multiple CD40 agonists are in early phase clinical trials for solid tumors, with each agonist exhibiting unique properties [161,162]. In CD40-expressing malignancies, such as chronic lymphocytic leukemia, CD40 monoclonal antibodies (mAbs) mediate direct tumor cell death via antibody-dependent cellular cytotoxicity [163,164].

### 3.5. NLRP3-Inflammasome

Inflammasomes are large cytosolic multiprotein complexes that mediate critical inflammatory innate immune responses in the host defense against microbial pathogens. The nucleotide-binding oligomerization domain-like receptors, or NOD-like receptors (NLRs) are a diverse family of intracellular PRRs. The nucleotide-binding domain and leucine-rich repeat family pyrin domain (NLRP) is a subfamily of NLRs and an important component of the inflammasome. The most well-characterized NLRP is NLRP3, which is expressed in macrophages, DCs and lymphocytes, in addition to non-immune populations, such as epithelial cells [165]. NLRP3 inflammasome activation is driven by the recognition of PAMPs or DAMPs generated in response to cellular stress. Examples include ATP, extracellular glucose and reactive oxidative species (ROS) [166,167,168,169,170]. When activated, NLRP3 forms the NLRP3 inflammasome by a multistep sequential process of priming and activation [166]. NLRP3 oligomerizes and cleaves procaspase-1 to caspase-1 and culminates in the release and proteolysis of the cytokines IL-18 and IL-1β. Gasdermin-D, a critical component of the inflammasome complex, permeabilizes the cell membrane and facilitates pyroptosis, as opposed to apoptosis [171,172,173]. Inflammasome-induced pyroptosis is canonically a caspase-1-dependent, immunostimulatory form of PCD in which target cell cytoplasmic contents are released to induce inflammation [174].

The impact of the NLRP3 inflammasome activation on tumorigenesis is complex. Preclinical evaluation has demonstrated that NLRP3 inflammasome mediated IL-1β and IL-18 release results in IFN-γ production and CD8^+^ T cell-dependent tumor regression [175,176]. In a preclinical model of hepatocellular carcinoma (HCC), estrogen receptor signaling increased cancer cell death through NLRP3 inflammasome-initiated, caspase-1-dependent pyroptosis and inhibition of autophagy in cancer cells [177]. In a preclinical model of colorectal cancer liver metastases, mice deficient in NLRP3 inflammasome had increased metastatic growth due to impaired IL-18 signaling and maturation of hepatic NK cells [178]. In the clinical setting, analysis of HCC patient tissue samples revealed an association between low expression of NLRP3 inflammasome components and more advanced HCC [179].

Dysregulated NLRP3 inflammasome activation has also been observed to promote tumorigenesis in multiple solid tumor murine models. 3′methylcholanthrene (MCA)-induced sarcomas in mice deficient in NLRP3 had decreased tumor burden compared to wild-type mice, attributed to NLRP3-mediated suppression of NK cell immune surveillance [180]. In oral squamous cell carcinoma cell lines, NLRP3 inflammasome activation and IL-1β secretion were associated with tumor progression, metastases and infiltration of immune suppressive myeloid cells, such as TAMs and MDSCs into the TME [181,182,183]. In addition, recent studies have shown that NLRP3 inflammasome activation within tumor cells can drive resistance to anti-PD-1 checkpoint inhibitor treatment. In anti-PD-1 treated solid tumor murine models, activated CD8^+^ T cells induced NLRP3 inflammasome activation within tumor cells, which resulted in downstream Wnt5a-mediated CXCR2 ligand expression and MDSC recruitment into the tumor tissue. This effect was abrogated by genetic and pharmacologic inhibition of NLRP3 [184].

Therapeutic targeting of NLRP3 is an emerging strategy, and a rationale exists for both NLRP3 inflammasome activation and inhibition. The novel NLRP3 agonist BMS-986299 is being studied in a phase I clinical trial as monotherapy and in combination with nivolumab and ipilimumab in advanced solid tumors [NCT03444753]. Saponins, derived from tree bark, contain molecules with strong proinflammatory properties that can activate the NLRP3 inflammasome and are being investigated for use as a vaccine adjuvant [185,186]. Though preclinical evaluations of NLRP3 inhibitors are expanding, they have not yet entered clinical trial testing [187]. In the HNSCC murine model, a novel NLRP3 inhibitor MCC950 delayed tumor growth, reduced MDSCs, Tregs and TAMs, while also increasing T cell infiltration within the TME [188].

Several recent studies have explored the mechanistic relationship between pyroptosis core proteins in cancer. Decreased expression of a primary mediator of pyroptosis, gasdermin D, is linked to enhanced cancer cell proliferation in vitro and tumor growth in vivo, whereas increased gasdermin E expression enhanced drug sensitivity of tumor cells [189,190]. These findings have led to therapeutic interest in activating pyroptosis as an antitumor strategy [191].

### 3.6. Dendritic Cell Directed Strategies

DCs are a key component of the innate immune system’s antitumor response. Strategies targeting PRRs, such as TLRs and RIG-I agonism, mediate their antitumor effect through activation of DCs [24]. Other DC activating and mobilizing agents, such as FLT3 ligands, have shown preclinical promise and are being evaluated in clinical trials [NCT03789097] [192]. Alternatively, activation of DCs can be achieved by blunting suppressive programs, which has led to the exploration of STAT3 and indoleamine 2,3-dioxygenase (IDO) inhibitors, now in various phases of clinical evaluation [193,194]. An increasingly nuanced understanding of DC biology has led to clinical breakthroughs in many DC-based therapeutic strategies that have earned FDA approval or advanced in clinical trials for the treatment of solid cancers [24].

### 3.7. Adoptive DC Strategies

Ex vivo vaccination of DCs with tumor-associated antigens (TAAs) and personalized tumor-specific antigens (TSAs) have also been explored. In these methods, autologous DC subsets are isolated from blood, activated ex vivo and loaded with antigen and reintroduced into the patient. Such DC vaccination strategies have long been an area of interest, with upwards of 200 clinical trials exploring their potential. Sipuleucel-T is an FDA-approved autologous DC vaccine in which DCs are incubated ex vivo with a fusion protein of GM-CSF and prostate-specific antigen (PSA) to treat metastatic, castrate-resistant prostate cancer [195]. Novel methods for optimizing the adoptive transfer of autologous antigen-loaded DCs are currently being developed, including personalized antigen selection and ex vivo activation with adjuvants [196]. In addition, combination with ICI is currently being evaluated to help overcome hostile immunosuppressive TMEs.

### 3.8. TAM Directed Strategies

Multiple therapeutic strategies have been found to drive polarization of protumorigenic M2 TAMs towards antitumor M1 TAMs, which include activation of CD40, TLR3, TLR4, TLR7/8 and TLR9 [89,90,151,155,197], covered in prior sections. Here, we discuss additional strategies being pursued in clinical trials.

#### 3.8.1. Colony Stimulating Factor 1 (CSF-1)

Blockade of macrophage CSF-1 binding to its cognate receptor (CSF-1R) resulted in the depletion of M2 TAMs, enrichment in M1 TAMs and tumor control in murine preclinical models [198,199,200]. Synergy in combination with anti-PD-1 and anti-CTLA-4 therapy has also been demonstrated [201]. Clinical trials of antibodies and small molecule inhibitors targeting CSF-1R are ongoing as monotherapy, and in combination with chemotherapy and ICI [197,202,203].

#### 3.8.2. PI3K-γ Inhibition

PI3K activation via the p110γ isoform within macrophages drives polarization to a protumorigenic M2 phenotype [204]. Consistent with this observation, inhibition of PI3K-γ isoform has been demonstrated to promote M1 polarization of TAMs and increase proinflammatory cytokines, resulting in antitumor immune activation and suppression of tumor growth in multiple solid tumor cancer models [204,205,206]. PI3K-γ inhibitor IPI-549 is currently undergoing clinical development for the treatment of solid tumors [207].

#### 3.8.3. CD47- Signal-Recognition Protein Alpha (SIRPα)

Therapeutic targeting of CD47, a surface glycoprotein and “don’t eat me” signal expressed on immune and tumor cells, which interacts with SIRPα on macrophages to suppress phagocytosis, is also being investigated. CD47-SIRPα blockade using CD47 antibodies was shown to restore phagocytosis of tumor cells by TAMs in vitro, stimulate antigen-specific T cells and limit tumor growth in murine models [208,209]. Multiple CD47 antibodies are undergoing clinical development for the treatment of solid tumors [210,211].

#### 3.8.4. Dendritic Cell-Specific Intercellular Adhesion Molecule-3-Grabbing Non-Integrin (DC-SIGN)

DC-SIGN, also known as CD209, is a marker of immunosuppressive TAMs. DC-SIGN+ TAM infiltration in clinical specimens is associated with an increased proportion of immunosuppressive regulatory T cells (Treg) and exhausted CD8^+^ T cells (TIGIT^+^ LAG3^+^). Treatment of human muscle-invasive bladder cancer single cell suspensions with monoclonal antibody targeting DC-SIGN in combination with PD-1 blockade demonstrated heightened antitumor activity as compared to monotherapy, establishing a rationale for clinical development [212].

#### 3.8.5. Other Macrophage-Directed Strategies

Inhibition of Arginase-1 (Arg1) has resulted in reduced tumor growth in preclinical models, and inhibitors of this enzyme have entered clinical development for the treatment of solid tumors [213,214]. Therapeutic blockade of CCR5, the receptor of protumoral chemokine CCL5, resulted in M2 to M1 macrophage polarization and a reduction in T-reg trafficking in a patient-derived functional in vitro organotypic culture model of hepatic colorectal metastases. This strategy demonstrated some activity in a Phase I clinical trial in patients with hepatic metastases of refractory colorectal carcinoma [215]. Class IIa histone deacetylase inhibitor, TMP195, demonstrated tumor reduction in breast cancer murine models by increasing the abundance of M1 macrophages [216]. Finally, cancer stem cell-derived WNT paracrine signaling in an ovarian cancer model upregulated M2 macrophages, which was reversed by WNT knockdown, suggesting this pathway as a potential target for TAM polarization [217]. It is noteworthy that inhibition of CCL2, an important chemokine that recruits monocytes, demonstrated promising preclinical activity; however, minimal clinical benefit was observed in clinical trials [218,219,220].

### 3.9. NK Cell Directed Strategies

While therapeutic targeting of activating receptors on NK cells has received much interest, most approaches have yet to reach clinical trials.

#### 3.9.1. Natural-Killer Group 2, Member D (NKG2D) Ligands

The most extensively studied activating receptor is NKG2D and its ligands, major histocompatibility complex (MHC) class I chain-related protein A (MICA) and MHC class I chain-related protein B (MICB). MICA and MICB are polymorphic proteins that are induced upon cell stress, damage or transformation and act as a “kill me” signal through the NKG2D receptor expressed on cytotoxic lymphocytes [221]. Unfortunately, many cancers shed MICA and MICB through proteolytic cleavage via ADAM10, ADAM17 and MMP14 matrix metalloproteases [222]. To combat MICA and MICB proteolytic shedding, small molecular inhibitors of matrix metalloproteases have been recently developed [223]. Additional strategies to reduce MICA and MICB shedding include antibody targeting of the α3 domains of MICA/B. These antibodies were shown to result in a significant NK cell-mediated antitumor response in an immunocompetent mouse model [224]. Similarly, proteosome inhibitors, such as bortezomib, may prevent MICA/B degradation and augment NK cell targeting of tumor cells [225]. Cancer cells with NKG2D-resistant variants can emerge through immune editing and epigenetic mechanisms which alter NKG2D ligand expression [226,227]. Thus, histone deacetylase inhibitors or chemotherapies have been investigated as treatments to upregulate the membrane expression of NKG2D ligands [228,229].

#### 3.9.2. NK Cell Engagers (NKCEs)

Bi- and tri-specific antibody constructs can bind multiple antigens and redirect NK cells to proximity of tumor cells and trigger an immune response and tumor cell death [230]. Multifunctional NKCEs with a tri-specific engager targeting two activating receptors, NKp46 and CD16, on NK cells and tumor antigen on cancer cells were observed to mediate significant tumor control in solid tumor murine models without notable toxicity [231]. These studies establish preclinical rationale for the clinical development of these molecules.

#### 3.9.3. NKG2A Inhibition

NKG2A, an inhibitory immune checkpoint molecule expressed on NK cells, and a subset of α/β T cells, has recently emerged as a potential therapeutic target for cancer therapy [232,233,234]. Monalizumab, a first in class ICI that targets the NKG2A receptor and simultaneously activates both NK cells and CD8^+^ T cells, has recently advanced to phase III clinical trials for head and neck squamous carcinoma, signifying an important advancement in NK cell therapy [235].

#### 3.9.4. Adoptive NK Cell Strategies

Due to the potent physiologic role of NK cells in tumor immunosurveillance, adoptive NK therapy (ACT) strategies have been an area of active investigation. Most current trials use allogeneic NK cells which are isolated from peripheral blood, propagated ex vivo and reinfused after lymphodepleting chemotherapy [236]. Genetic engineering of activating receptors on NK cells, such as the NKG2D receptor, has been used to optimize antitumor activity. NK cell ACT has demonstrated robust control of early metastasis, tumor specificity and more favorable toxicity profiles, when compared to adoptive T cell strategies. However, concerns over efficacy in solid tumors for NK cell ACT exist, including limited proliferative capacity, ability to infiltrate the tumor and presence of local immunosuppressive mechanisms within the TME [237,238]. Methods to increase the persistence of NK cells after infusion are being investigated, such as HLA knockdown to prevent rejection by the recipient’s immune system [239]. Tumors can escape NK cell-driven cytotoxicity by secreting immunosuppressive factors, such as TGF-β and adenosine, increasing immunosuppressive tryptophan metabolites via upregulating IDO, shedding MICA and MICB proteins and recruiting suppressive populations, such as Tregs, MDSCs and M2 TAMs [240,241,242,243]. Therapies targeting these mechanisms may also serve to reinvigorate a suppressed NK cell compartment.

## 4. Conclusions and Future Directions

There have been significant advances made in our understanding of the innate immune system’s contribution and response to tumorigenesis. Rational immuno-oncology combination strategies to modulate these pathways and activate both innate and adaptive immunity carry the potential to improve immunotherapy outcomes in all patients, particularly those with immunologically “cold” tumors.

Enthusiasm over novel innate immune strategies must be met with appropriate consideration regarding the risk of immune-related adverse events (irAEs). By virtue of their desired antigen-independent on-target effects to stimulate highly potent and conserved proinflammatory mechanisms, the risk of off-target toxicity is not insignificant. As clinical use expands, it will be important to distinguish irAEs specific to innate immune investigational agents from potentiation of irAEs caused by ICI, as many investigational strategies involve combination with ICI. There is an unmet clinical need to develop novel evidence-based protocols for toxicity management, acknowledging the expanding role of combinatorial approaches.

Despite considerable scientific and clinical progress, there remain gaps in our understanding of the dynamic interplay between cancer and immune cells within the TME following therapeutic intervention. Of critical importance will be to further define context-dependent roles of innate immune pathways in different tumor and genomic subtypes. For example, activation of innate immunity can be a “double-edged sword” and have pro and antitumorigenic roles in tumor development and progression, depending on tumor type, genetic/epigenetic, metabolic and microenvironmental context. However, the selective context-specific targeting of the innate immune system has the potential to become a cornerstone of immunotherapy strategies for the treatment of solid tumors.

## Figures and Tables

**Figure 1 vaccines-09-00138-f001:**
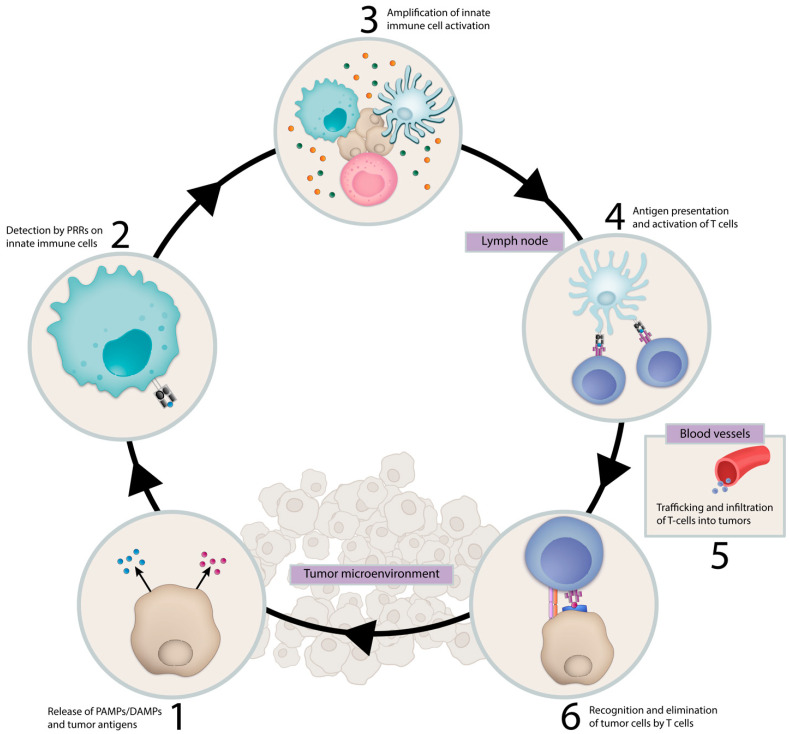
Cancer Immunity Cycle. Innate immune cells facilitate the immune system’s response against a recognized pathogen. This process is initiated by detection of pathogen-associated molecular patterns (PAMPs) and damage-associated molecular patterns (DAMPs) and other unique tumor antigens by innate immune cells which result in antigen presentation and activation of antigen-specific T cells in tumor draining lymph nodes. These T cells traffic to the tumor and mediate tumor elimination. Abbreviations: PAMPs, pathogen-associated molecular patterns; DAMPs, damage-associated molecular patterns; PRRs, pattern-recognizing-receptors.

**Figure 2 vaccines-09-00138-f002:**
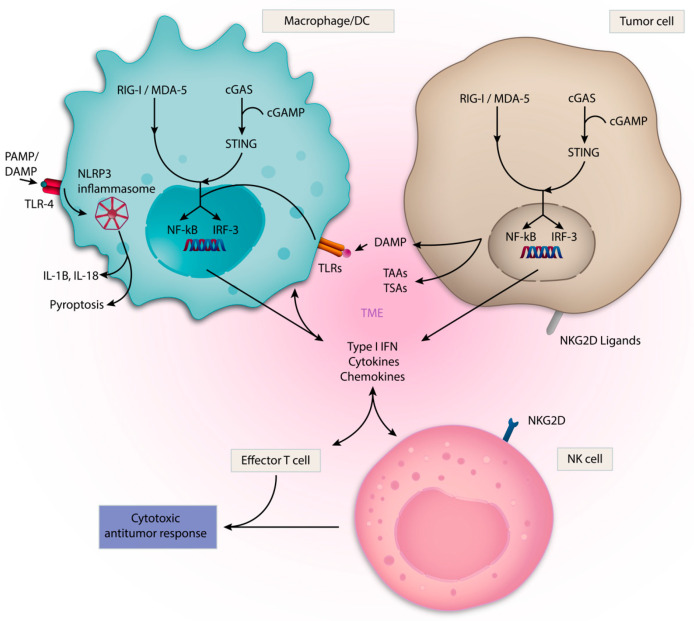
Innate Immune sensing pathways. A diverse set of pattern recognition receptors (PRRs) detect damage-associated molecular patterns (DAMPs), pathogen-associated molecular patterns (PAMPs) or tumor-associated antigens (TAAs) and activate innate immune cells within the TME. These sensors can also be expressed by tumor cells themselves. Multiple PRRs, including cGAS/STING, RIG-I and TLRs, promote transcription of proinflammatory genes via IRF-3 and NF-kb. These processes result in Type I IFN, cytokine and chemokine production, which supports a cytotoxic antitumor response mediated by effector T cells and NK cells. The NLRP3 inflammasome drives a proinflammatory response through IL-1β and IL-18, in addition to mediating pyroptosis. NKG2D ligands activate NK cells.

**Figure 3 vaccines-09-00138-f003:**
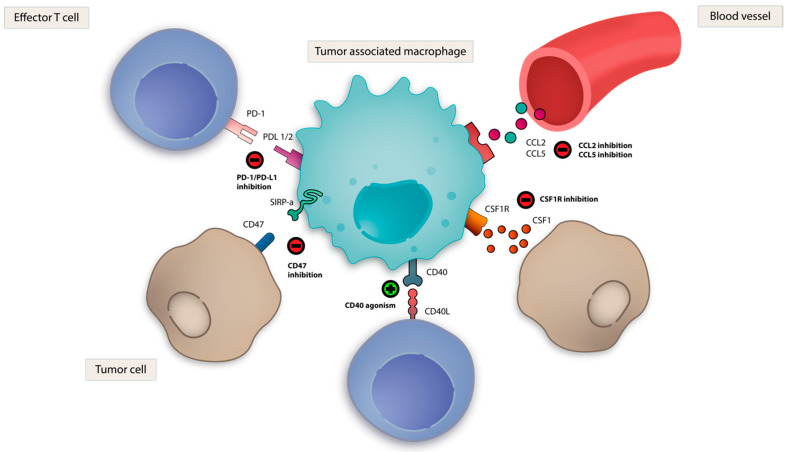
Strategies targeting TAMs. Multiple therapeutic strategies exist to target tumor-associated macrophages (TAMs) in cancer immunotherapy. CSF-1R inhibition and CD40 agonism promote TAM polarization to proinflammatory M1 phenotype. CD47-SIRPα blockade restores TAM-mediated phagocytosis. PD-1 blockade reverses TAM-mediated immunosuppression. CCL5 and CCL2 blockade attenuate TAM recruitment.

**Table 1 vaccines-09-00138-t001:** Innate immune cell expression of PRRs and PAMP/DAMP-sensing pathways.

	DC	TAM	Mo-MDSC	PMN-MDSC	Neutrophils	NK Cells	Basophils	Eosinophils	Mast Cells
TLR7/8									
TLR9									
RIG-I									
cGAS-STING									
NLRP3									
CSF-1R									
CD40									
NKG2D									

Green = expression of innate immune pathway or receptor is observed in human cell types based on literature cited in main body of text. This is a conceptual and binary illustration which does not reflect contextual and dynamic nuances in expression.

**Table 2 vaccines-09-00138-t002:** Summary of agents in clinical development.

PRR	Agent	Molecule Type	Route of Administration	Cancer Type(s)	Clinical Phase of Development	
TLR8	VTX-2337 (motilomod)	Small molecule	Intratumoral	ovarian, HNSCC	I, II	
dTLR7/8	NKTR-262	Small molecule	Intratumoral	Solid tumors	I, II	
TLR9	SD-101	CpG-C class ODN	Intratumoral	Solid tumors	I, II	
	EMD 1201081	Synthetic ODN	Subcutaneous Injection	HNSCC	I, II	
	CPG 7909	CpG ODN	Subcutaneous Injection	Lymphomas	I, II	
	IMO-2125 (Tilsotolimod)	Synthetic ODN	Intratumoral	melanoma, Solid tumors	I, II	
	CMP-001	CpG ODN	Intratumoral	Solid tumors	I, II	
RIG-I	SLR-14	Synthetic stem loop RNA	Intratumoral	Solid tumors	Pre-clinical	
	RGT-100 (MK-4621)	Synthetic oligonucleotide	Intratumoral	Solid tumors		
MDA-5	BO-112 (poly(I:C))	Synthetic dsRNA	Intratumoral	Solid tumors	I	
STING	E7766	Novel macrocycle-bridged	Intravenous	Solid tumors	I	
	GSK3745417	Small molecule	Intravenous	Solid tumors	I	
	MIW815 (ADU-S100)	Synthetic CDN	Intratumoral	Solid tumors	I, II	
	MK1454	Small molecule	Intratumoral	Solid tumors	I, II	
	BMS-986301	Small molecule	Intratumoral *	Solid tumors	I	
NLRP3	BMS-986299	First in class agonist *	Intratumoral *		I	
**Other innate immune targets**						
CSF-1R	Cabiralizumab	Monoclonal antibody	Intravenous	Solid tumors	I, II	
	JNJ-40346527	Monoclonal antibody	Intravenous	Advanced prostate cancer	I, II	
	PLX3397	Small molecule	Oral	Solid tumors	I, II	
	MCS110			Solid tumors	I, II	
	IMC-CS4	Monoclonal antibody	Intravenous	Solid tumors	I	
CD40	APX005M	Monoclonal antibody	Intravenous	Solid tumors	I, II	
	CP-870,893	Monoclonal antibody	Intravenous	Solid tumors	I	
	Selicrelumab	Monoclonal antibody	Intravenous	Solid tumors	I, II	
PI3K (delta) Inhibitors	IPI-549	Small molecule	Oral		Solid tumors	I, II
Class IIa histone deacetylase inhibitor	TMP-195	Small molecule	Oral		Solid Tumors	Preclinical
IDO inhibitors	Indoximod	Small molecule	Oral		Solid tumors	I, II
STAT3 inhibitors	Siltuximab	Monoclonal antibody	Intravenous		Solid tumors	I, II
	WP1066	Small molecule	Oral		Solid tumors	I
	TT-101	Small molecule	Oral		Solid tumors	I

* BMS-986301 is being evaluated for systemic intramuscular administration. The novel NLRP3 agonist BMS-986299 is being studied in a phase I clinical trial as mon-otherapy and in combination with nivolumab and ipilimumab in advanced solid tumors [NCT03444753].

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
