# Peer review of "Targeting Innate Immunity in Cancer Therapy"

_vaccines, 2021, doi:10.3390/vaccines9020138_

Round 1
Reviewer 1 Report
Srikrishman R et al. describe the current therapies for targetting cancers via components of the innate immune system including molecular pathways (TLRs, CD40, RLRs etc) and the main cell lineages (NK cells, TAMs) before finishing with oncolytic viruses. Generally speaking the review is well written, although they paint with a broad brush, and the figures/table re-enforce and explain the relationships between mechanisms.
One aspect of the introduction that would benefit from a minor expansion is the role and perception of hot vs cold tumours. Depending on the tumour type, the infiltration of leucocytes can indicate a good or poor prognosis, and the type of infiltrating cell may also influence survival.
There is sufficient literature on the anti-neoplastic potential of the CSF-1R inhibitor GW2850 to warrant its inclusion.
In section 7 the authors list therapeutic strategies from Ln 347 onwards (firstly, secondly etc) but the numbering jumps from third to fifth, with no fourth.
At Ln 371 the authors mention CCL5 blockade but then discuss ‘a receptor’ which is confusing. It is not clear what is meant. Furthermore, it is suggested that the relationship between CCL5 and Treg homing to tumours is highlighted (e.g. PMID: 19155524).
The section on oncolytic viruses is interesting but doesn’t fit with the rest of the review. The innate immune system is involved but in much the same way as other cytotoxic treatments. It is suggested that it could be omitted and used in another manuscript.
The authors mention the role of MDSC but do not expand upon them in a larger section and also do not mention in detail other myeloid cells (monocytes or granulocytes) that participate in tumorigenesis. It is suggested that the authors add a few sentences on both MDSC and monocytes in the TAM section, including any differential anti-tumour activity of monocyte subsets. There is not sufficient space to fully explore the subject of macrophage heterogeneity, which would be a review of its own, but it would help the reader if the authors could expand slightly the current breadth of the subject rather than just referring to refs [130-132] at Ln 334. For example, recent scRNAseq studies.
Sections 6, 7 and 8 would benefit from a subheading to break up the text and signpost the author’s narrative for the reader.
The authors should consider expanding the Abstract to include some of the keywords and ensuring the words present in the Abstract are repeated in the title for search engine optimisation. Currently the Abstract is somewhat vague.
Author Response
Reviewer 1
- One aspect of the introduction that would benefit from a minor expansion is the role and perception of hot vs cold tumours.
RESPONSE: We thank the reviewer for this comment. We have addressed this on Page 2, 2nd paragraph of the Introduction by restructuring the paragraph to include discussion of “hot” and “cold” tumors and added appropriate references.
- Depending on the tumour type, the infiltration of leucocytes can indicate a good or poor prognosis, and the type of infiltrating cell may also influence survival.
RESPONSE: We have addressed this in Page 2, 2nd paragraph of the Introduction.
- There is sufficient literature on the anti-neoplastic potential of the CSF-1R inhibitor GW2850 to warrant its inclusion.
RESPONSE: We agree with this comment and have added the following reference on GW-2580, Escamilla et al. CSF1 receptor targeting in prostate cancer reverses macrophage-mediated resistance to androgen blockade therapy; Cancer Res 2015 Mar 15;75(6):950-62. doi: 10.1158/0008-5472.CAN-14-0992. There are no ongoing clinical trials of GW-2580, so it was not added to the Table. - In section 7 the authors list therapeutic strategies from Ln 347 onwards (firstly, secondly etc) but the numbering jumps from third to fifth, with no fourth.
RESPONSE: We have addressed this comment on Page 16-17, within the context of restructuring the section based on other reviewers’ comments. - At Ln 371 the authors mention CCL5 blockade but then discuss ‘a receptor’ which is confusing. It is not clear what is meant. Furthermore, it is suggested that the relationship between CCL5 and Treg homing to tumours is highlighted (e.g. PMIDî‚’ 19155524î‚‚ )
RESPONSE: We thank the reviewer for this comment and have addressed the text appropriately on the first paragraph of Page 18 and have inserted the suggested reference.
- The section on oncolytic viruses is interesting but doesn’t fit with the rest of the review. The innate immune system is involved but in much the same way as other cytotoxic treatments. It is suggested that it could be omitted and used in another manuscript.
RESPONSE: We agree with this comment and have removed the section on oncolytic viruses. - The authors mention the role of MDSC but do not expand upon them in a larger section and also do not mention in detail other myeloid cells (monocytes or granulocytes) that participate in tumorigenesis. It is suggested that the authors add a few sentences on both MDSC and monocytes in the TAM section, including any differential anti-tumour activity of monocyte subsets.
RESPONSE: We thank the reviewer for this feedback with regards to content. Based on other reviewers’ comments, we have added an independent section on MDSCs found on Page 5 of the manuscript. - There is not sufficient space to fully explore the subject of macrophage heterogeneity, which would be a review of its own, but it would help the reader if the authors could expand slightly the current breadth of the subject rather than just referring to refs 130, 132 at Ln 334. For example, recent scRNAseq studies
RESPONSE: We thank the reviewer for this comment and have added a few sentences and two references to this effect on Paragraph 1 on Page 4.
- Sections 6, 7 and 8 would benefit from a subheading to break up the text and signpost the author’s narrative for the reader.
RESPONSE: We have restructured Sections 6, 7, and 8, such that cell type immune biology and therapeutics have been subdivided into distinct sections.
The authors should consider expanding the Abstract to include some of the keywords and ensuring the words present in the Abstract are repeated in the title for search engine optimisation. Currently the Abstract is somewhat vague.
RESPONSE: We thank the reviewer for this comment. We have expanded the abstract accordingly.
Reviewer 2 Report
The manuscript by Rameshbabu et al. reviewed how targeting different innate immune pathways can facilitate the therapy of different cancers. The review is nicely organized, thorough, and exhaustive.
However, there are a few issues that authors need to address.
- The authors started by enumerating how manipulation of different innate pathways can affect cancer progression. At the very end, the authors discussed the roles of macrophages and NK cells. The problem with this organization is that it is difficult for the reader to find information about any specific cell type in one place. For example, the authors elucidate several studies and mechanisms explaining how targeting different molecules/ pathways in dendritic cells affects the treatment/ progression of different cancers. Yet, it will be very difficult for a reader to find how targeting DCs as a whole affects different cancers. Same thing about MDSCs. All the information is there, but because of the organization of the manuscript, difficult to find. The authors should summarize their findings in a table (or multiple tables, one for each pathway) with columns such as drug name, pathway targeted, mechanism of action, cancer type studied, cell type involved, dose and routes, references, etc. The current table is insufficient and doesn't do justice to the wealth of information the authors discussed throughout the manuscript.
- It is surprising that such a nicely organized manuscript didn’t discuss the roles of neutrophils and mast cells in the context of cancers. These should be discussed separately, if space permits, or at least mentioned in the tables when relevant pathways are mentioned,
Overall, the authors should be commended for writing a comprehensive review on a topic that is becoming increasingly important.
Author Response
The review is nicely organized, thorough, and exhaustive. To address:
- The authors started by enumerating how manipulation of different innate pathways can affect cancer progression. At the very end, the authors discussed the roles of macrophages and NK cells. The problem with this organization is that it is difficult for the reader to find information about any specific cell type in one place. For example, the authors elucidate several studies and mechanisms explaining how targeting different molecules/pathways in dendritic cells affects the treatment/ progression of different cancers. Yet, it will be very difficult for a reader to find how targeting DCs as a whole affects different cancers. Same thing about MDSCs. All the information is there, but because of the organization of the manuscript, difficult to find. The authors should summarize their findings in a table (or multiple tables, one for each pathway) with columns such as drug name, pathway targeted, mechanism of action, cancer type studied, cell type involved, dose and routes, references, etc. The current table is insufficient and doesn't do justice to the wealth of information the authors discussed throughout the manuscript.
RESPONSE: We appreciate this comment. We have made a new Table which indicates which pathways are active in each cell type. We have found it difficult to address the entirety of this comment with regards to exhaustive inclusion of the effects of each drug on each cell type by cancer type as these cell types have been studied in a diverse number of settings. We hope the new table will suffice in that it gives the reader a sense of what cell type is being manipulated by a given treatment. As noted above, we have reorganized the manuscript to make the information more easily trackable throughout the manuscript.
- It is surprising that such a nicely organized manuscript didn’t discuss the roles of neutrophils and mast cells in the context of cancers. These should be discussed separately, if space permits, or at least mentioned in the tables when relevant pathways are mentioned
RESPONSE: We agree with this reviewers comment and have added a section dedicated to neutrophils and mast cells found on page 4 and 5, respectively.
Reviewer 3 Report
Thanks for giving me the opportunity for reviewing this manuscript entitled “Targeting Innate Immunity in Cancer Therapy”. Although adaptive immunity plays critical role for anti-tumor immunity, targeting innate immunity has many therapeutic potentials to modulate the immunosuppressive tumor microenvironment. This manuscript described an important aspect.
Major points
#1 Authors should consider to change the order of sections. The current order is: 1. Introduction (Cancer Immunity Cycles and innate immunity), 2-6: Innate receptors, 7 macrophages, 8 NK cells, 9, Oncolytic virus. This flow is not easy for readers to understand.
Instead, I would suggest the following flow.
- Introduction,
- Key players in innate immunity (macrophages and NK cells) including their activation and immune regulation.
- Discussion on how we can improve innate anti-tumor immunity by PRRs, agonists, oncolytic virus etc.
#2 Figure 2. TLR9 is described on the surface of TAMs. TLR9 is mainly expressed on endosome, although some studies show cell-surface expression on certain immune subsets. Given that this manuscripts describes other TLR agonists, this Figure should be described as TLRs, rather than TLR9.
The interaction between NK cells and DCs should be bi-directional. This should be clarified in the Figure.
#3. While stimulation of innate immunity is a possible approach, toxicity remains a major concern. Potential Immune-related adverse events should be discussed to some degree in Conclusion section.
Other points
- Line229: CD40 section. Toxicity of CD40 agonist should be mentioned.
- Line 282: Pyroptosis is a caspase-1 dependent, immunostimulatory form….As authors described later, Caspase-1-indpendent pyroptosis pathway exists (Casp3-GSDME).
- Line287 caspase1…should be caspase-1
- Line 97, 144, 199, 234, 254, 272. Dendritic cells can be described as DCs, as authors use the abbreviation in Line 20. Other abbreviations have to be checked.
Author Response
Reviewer 3
- Authors should consider to change the order of sections. The current order is: 1. Introduction (Cancer Immunity Cycles and innate immunity), 2-6: Innate receptors, 7 macrophages, 8 NK cells, 9, Oncolytic virus. This flow is not easy for readers to understand.
RESPONSE: We appreciate the reviewers opinion on the layout of information in the manuscript. To address this, we have created independent sections similar to suggestion proposed: including key players in innate immunity section with independent discussion of each cell type; followed by therapeutic strategies as it pertains to the major pathways and cell types.
- Figure 2. TLR9 is described on the surface of TAMs. TLR9 is mainly expressed on endosome, although some studies show cell-surface expression on certain immune subsets. Given that this manuscripts describes other TLR agonists, this Figure should be described as TLRs, rather than TLR9.
RESPONSE: We appreciate the comment and have addressed the figure accordingly.
- The interaction between NK cells and DCs should be bi-directional. This should be clarified in the Figure.
RESPONSE: We appreciate the comment and have addressed the figure accordingly.
- While stimulation of innate immunity is a possible approach, toxicity remains a major concern. Potential Immune-related adverse events should be discussed to some degree in Conclusion section.
RESPONSE: We appreciate the comment and have addressed this by including a paragraph discussing IRAE in Paragraph 2 of the Conclusion section on Page 19.
Other points
- Line229: CD40 section. Toxicity of CD40 agonist should be mentioned.
- Line 282: Pyroptosis is a caspase-1 dependent, immunostimulatory form….As authors described later, Caspase-1-indpendent pyroptosis pathway exists (Casp3-GSDME).
- Line287 caspase1…should be caspase-1
- Line 97, 144, 199, 234, 254, 272. Dendritic cells can be described as DCs, as authors use the abbreviation in Line 20. Other abbreviations have to be checked.
RESPONSE:
Thank you for the comments, we have addressed them as follows:
- In the second paragraph of the CD40 section, we have briefly discussed toxicity associated with CD40 agonism.
- We have clarified that "NLRP3 driven" pyroptosis is canonically Caspase-1 driven.
- We have addressed all other outstanding edits.
Round 2
Reviewer 3 Report
AUthors have addressed concenrs, and improved the quality. I would suggest this for publication.